# Single-Domain Antibodies as Therapeutics for Respiratory RNA Virus Infections

**DOI:** 10.3390/v14061162

**Published:** 2022-05-27

**Authors:** Keke Huang, Tianlei Ying, Yanling Wu

**Affiliations:** 1MOE/NHC Key Laboratory of Medical Molecular Virology, School of Basic Medical Sciences, Shanghai Medical College, Fudan University, Shanghai 200032, China; 16301010028@fudan.edu.cn; 2Shanghai Engineering Research Center for Synthetic Immunology, Shanghai 200032, China

**Keywords:** single-domain antibody, nanobody, respiratory RNA virus, antiviral therapeutics, inhalable property

## Abstract

Over the years, infectious diseases with high morbidity and mortality disrupted human healthcare systems and devastated economies globally. Respiratory viruses, especially emerging or re-emerging RNA viruses, including influenza and human coronavirus, are the main pathogens of acute respiratory diseases that cause epidemics or even global pandemics. Importantly, due to the rapid mutation of viruses, there are few effective drugs and vaccines for the treatment and prevention of these RNA virus infections. Of note, a class of antibodies derived from camelid and shark, named nanobody or single-domain antibody (sdAb), was characterized by smaller size, lower production costs, more accessible binding epitopes, and inhalable properties, which have advantages in the treatment of respiratory diseases compared to conventional antibodies. Currently, a number of sdAbs have been developed against various respiratory RNA viruses and demonstrated potent therapeutic efficacy in mouse models. Here, we review the current status of the development of antiviral sdAb and discuss their potential as therapeutics for respiratory RNA viral diseases.

## 1. Introduction

Respiratory diseases caused by various types of virus infections have been the focus of global health concerns and are one of the leading causes of death in developing countries [1]. According to the nucleic acid types, respiratory viruses can be divided into RNA and DNA viruses. However, the primary viruses causing the epidemics of respiratory infections in the last two decades were RNA viruses, such as the severe acute respiratory syndrome coronavirus (SARS-CoV) in 2003, the influenza H1N1 virus in 2009, the Middle East respiratory syndrome coronavirus (MERS-CoV) in 2012, and the SARS-CoV-2 in 2019 [2]. Therefore, the development of effective therapeutics for respiratory RNA viruses is rival to combat infectious diseases.

Respiratory RNA viruses include coronaviruses (SARS-CoV-1, MERS-CoV, and SARS-CoV-2), influenza viruses, respiratory syncytial virus (RSV), and others [3,4]. With the development of innovative recombinant DNA technologies, monoclonal antibodies (mAbs) have been proven effective in controlling respiratory RNA viral diseases. In 1998, the mAb palivizumab targeting RSV fusion (F) protein was approved by the FDA as prophylaxis against serious lower respiratory tract disease caused by RSV in children at high risk [5]. However, the high production costs of mAbs limit their commercial market, and the large size of mAbs leads to their low tissue accessibility and penetration, thus affecting their therapeutic efficacy. These features obstruct the development of mAbs [6]. Single-domain antibodies (sdAbs), consisting of only variable domains, have many advantages compared to mAbs. Their smaller size enables tissue penetration, so they could recognize epitopes that are normally not accessible for mAbs. In addition, the smaller size and higher stability of sdAbs make administration by inhalation possible, which is more suitable for treating respiratory diseases. It is also easy to express sdAbs in bacteria so that the production costs could be reduced. Therefore, sdAbs are a promising alternative to conventional mAbs [7,8]. In this review, we summarize the development of sdAb-based therapeutics for respiratory RNA virus infections and the strategies of antigen-specific sdAb screening.

## 2. Single-Domain Antibodies

In 1993, Hamers-Casterman et al. found that camelids could produce homodimeric heavy chain-only antibodies (HCAbs) devoid of light chains and the first constant domain (CH1) [9]. Two years later, Greenberg et al. reported that sharks and other cartilaginous fish could produce a type of HCAbs called Ig new antigen receptors (IgNARs) [10]. IgNARs compose of two identical heavy chains, each comprising five constant domains and a variable domain named V_NAR_ that is responsible for antigen recognition [11]. The autonomous variable domains of HCAbs and IgNARs are called sdAbs, also known as nanobodies or VHHs if coming from the camelid family (including camels, llamas, and vicugna) [12]. Compared to cartilaginous fish, camelids are easier to access and can generate stronger sdAbs after antigen immunization, and VHHs share relatively higher homology with human immunoglobulin heavy chain variable region (IGHV) genes [11,13]. Therefore, VHHs attracted more interest than V_NAR_ in the biological drug field.

There are many strategies to identify sdAbs targeting specific antigens, including immunization of camelid and transgenic mice or panning by phage/yeast library with human sdAbs, camelid nanobodies, and IgNARs (Figure 1). The most popular strategy is that VHH genes were cloned from the peripheral blood lymphocytes of camelids immunized with specific antigens, followed by constructing a nanobody library by phage display and isolating nanobodies from this library. For example, Detalle et al. identified a monovalent RSV F protein-specific nanobody from llamas that received repetitive immunization with soluble recombinant F protein [14]. In another study, Xu et al. created mice (named nanomice) that could produce high-affinity nanobodies by inserting a VHH cassette instead of the VH locus in mouse embryonic stem cells. Then they immunized these nanomice with the receptor-binding domain (RBD) and the stabilized prefusion spike (S) protein of SARS-CoV-2 and isolated two nanobodies by phage display [15]. However, the sequences of VHHs with nonhuman origin may increase the immunogenicity risk in humans, leading to the need for humanization in VHH development. Wu et al. identified human VHs exhibiting biophysical properties very similar to VHHs from a fully human sdAb library constructed by using the germline 3-66*01 VH framework regions, indicating the potential of human sdAbs as alternatives to VHHs [16,17].

As the smallest antibodies, sdAbs have several beneficial characteristics, such as low molecular weight (12–15 kDa), small size (4 × 2.5 nm) [18], as well as high stability and solubility due to their longer CDR3 loops than conventional antibodies [19,20]. Because of the small size, sdAbs can recognize cavities or hidden epitopes that are not accessible to conventional antibodies [21]. For example, Wu et al. identified several fully human sdAbs that recognize a cryptic epitope located in the SARS-CoV-2 spike trimeric interface and have potent neutralization [16]. Moreover, sdAbs can be easily produced in bacteria, greatly reducing the production cost.

In addition, sdAbs can be easily engineered to be multivalent or multispecific, improving their binding affinity and breath [22] and prolonging half-life in vivo [23]. Lauren et al. reported four sdAbs, designed as SD36, SD38, SD83, and SD84, which could bind to highly conserved epitopes of the influenza A and B virus hemagglutinins (HAs), respectively. By fusing these sdAbs with peptide linkers, two multidomain antibodies (MDAbs) were generated and demonstrated the improved binding breadth and neutralizing potency of individual sdAbs [24]. In another study, a bispecific sdAbs bn03 was constructed by linking two sdAbs (n3113v and n3130v) that recognized two different conserved epitopes on SARS-CoV-2 RBD and exhibited potent neutralizing activity to all SARS-CoV-2 variants including Omicron [25]. Importantly, inhalation of bn03 significantly reduced viral titer in the lung in mild or severe SARS-CoV-2 infectious mice [25], indicating inhalation as a favorable route for delivering human sdAbs.

Currently, sdAbs have been developed to treat and diagnose various diseases [18]. In 2019, FDA approved Sanofi’s caplacizumab (an anti-von Willebrand factor nanobody and the first nanobody approved by FDA) for treating acquired thrombotic thrombocytopenic purpura (aTTP), a rare disease characterized by excessive blood clotting in small blood vessels [26]. A nanobody targeting human epidermal growth factor receptor 2 (HER2) is being evaluated in detecting breast-to-brain metastasis by Positron Emission Tomography (PET)/Computed Tomography (CT) imaging in phase II clinical trials (NCT03331601) [27,28]. Furthermore, due to the small size and favorable biophysical characteristics, some studies have reported that directly delivering nanobodies into the lung can block the viral invasion of airway epithelial cells in situ by inhalation [25,29,30]. Therefore, nanobodies can be administered by inhaled delivery and are particularly suitable for treating respiratory diseases caused by respiratory RNA virus infections.

## 3. The Mechanisms of Single-Domain Antibodies Inhibiting Respiratory RNA Virus Infections

sdAbs can neutralize viruses by several different mechanisms in parallel with the virus life cycle (Figure 2) [31]. Respiratory RNA viruses can be classified into enveloped and non-enveloped viruses based on the presence or absence of a lipid membrane. Except for human rhinoviruses (HRVs), all other respiratory RNA viruses are enveloped viruses (Table 1) [32,33,34,35,36,37,38]. The main life-cycle steps of enveloped and non-enveloped viruses are similar, beginning with the entry into the host cell and ending with the release of virion progeny from the host cell. The majority of neutralizing sdAbs could serve as therapeutics for respiratory RNA virus infections by inhibiting the virion entry, while a few antiviral sdAbs can inhibit the viral genomic replication and the release of virion progeny [31].

The process of viral entry usually starts from the attachment to a host cell; hence sdAbs can block viral entry by directly interfering with the interactions between the virus and host receptors, such as the RBD of SARS-CoV-1 and SARS-CoV-2 S proteins [39], and the receptor-binding site of influenza HA [40]. For example, a number of sdAbs targeting the RBD of SARS-CoV-2 S protein can block the interaction of RBD with the receptor angiotensin-converting enzyme 2 (ACE2) [41]. After attachment, the fusion of viral and host membranes is triggered and the viral genome is released into the cell cytoplasm from the endosome. Therefore, sdAbs can also block viral entry by binding to viral proteins that mediate membrane fusion and the release of the viral genome, such as the F protein of RSV [14], the HA and M2 of influenza viruses [42,43], and the S protein of SARS-CoV-2 [44]. Previous studies have reported that one sdAb targeting the non-ACE2 binding site of RBD can neutralize SARS-CoV-2 by inhibiting the conformational change of S protein that is essential for membrane fusion [45]. Moreover, some sdAbs can inhibit viral replication by binding to the viral proteins that are essential for virion release from the host cell, such as the neuraminidase (NA) of influenza viruses [46]. Apart from binding to the viral surface proteins, sdAbs can also inhibit the viral replication by binding to the viral intracellular proteins, such as the nucleoprotein of influenza viruses that mediates the viral nuclear trafficking and packaging [43,47].

## 4. Coronaviruses

Coronaviruses (CoVs) are enveloped viruses with a positive-sense RNA genome that usually cause mild to moderate respiratory diseases in humans, which lead to several outbreaks of respiratory diseases. Based on their phylogenetic relationships and genomic structures, CoVs are classified into four genera (*Alphacoronavirus*, *Betacoronavirus*, *Gammacoronavirus*, and *Deltacoronavirus*). Among them, the highly pathogenic CoVs belong to betacoronaviruses and cause acute lung injury (ALI)/acute respiratory distress syndrome (ARDS) in humans, such as SARS-CoV-1, MERS-CoV, and SARS-CoV-2 [48]. The CoV virion contains at least four structural proteins: spike protein (S), nucleocapsid protein (N), membrane protein (M), and envelope protein (E) [49]. S protein is a class I viral fusion protein that mediates the process of viral entry by binding to the target host cell and initiating fusion with the cell membrane. S is homotrimeric, with each subunit consisting of two domains, S1 and S2. S1 contains an RBD that binds to the receptor of the host cell. Different CoVs enter target cells with distinct receptors. For example, SARS-CoV-1 and SARS-CoV-2 bind to ACE2 for host cell entry, whereas MERS-CoV recognizes dipeptidyl peptidase 4 (DPP4) as its receptor. S2 mediates the subsequent membrane fusion to enable entry to the host cytoplasm [50,51]. Due to these essential roles in viral entry, the S protein is the main target of neutralizing sdAbs.

In 2003, the emergence of SARS-CoV-1 has caused a devastating pandemic. Wrapp et al. reported that two potently neutralizing sdAbs, VHH-72 and VHH-55, were isolated from a llama immunized with prefusion-stabilized SARS-CoV-1 and MERS-CoV spikes. These sdAbs are bound to the SARS-CoV-1 and MERS-CoV RBDs with a high affinity of 1.15 nM and 0.079 nM, respectively (Table 2). In addition, VHH-72 is also bound to SARS-CoV-2 RBD with an affinity of 38.6 nM. However, VHH-72 must be fused with an Fc domain of human IgG1 with a (GGGGS)_3_ linker to neutralize SARS-CoV-2 pseudovirus with an IC_50_ of ~0.2 μg/mL [39]. Besides, several sdAbs binding to SARS-CoV-1 RBD were also identified, such as 5F8 [52], Fu2 [53], S1-1, S1-RBD-6 [54], and S14 [55] (Table 2).

MERS-CoV was first identified in Saudi Arabia in 2012 with a 35% mortality rate, causing persistent endemics in Middle East regions. Raj et al. constructed a complementary DNA library from bone marrow cells of two dromedary camels immunized with a modified vaccinia virus encoding the MERS-CoV S protein and subsequently challenged with live MERS-CoV. From this library, four VHHs (VHH-1, VHH-4, VHH-83, and VHH-101) were identified with high binding affinity to MERS-CoV S protein and displayed neutralizing activity. VHH-83 had the prophylactic efficacy in the K18 transgenic mouse model infected with a lethal dose of MERS-CoV [56] (Table 2). However, none of them bound to the D539N variant located in RBD, suggesting that VHHs neutralized MERS-CoV most likely by blocking binding to DPP4. NbMS10 was selected from a VHH library by phage display and bound to MERS-CoV RBD-Fc with a high affinity of 0.87 nM. The sdAb neutralized MERS-CoV by blocking the binding between MERS-CoV RBD and DPP4 and the IC_50_ of NbMS10 was 3.52 μg/mL. Meanwhile, the dimeric and trimeric NbMS10, generated by linking monomeric NbMS10 with a GGGGS linker, showed higher binding affinity to RBD and cross-neutralizing activity against divergent strains of MERS-CoV. In addition, NbMS10-Fc had potently prophylactic and therapeutic efficacy in hDPP4-Tg mice challenged with lethal MERS-CoV [57,90] (Table 2).

The current COVID-19 pandemic caused by SARS-CoV-2 has been persistent for more than two years, posing a significant threat to human health [36]. Recently, more than 600 sdAbs have been developed for the therapeutics of SARS-CoV-2 (http://opig.stats.ox.ac.uk/webapps/covabdab/, accessed on 4 May 2022). Medium-to-high titers of cross-neutralizing antibodies against SARS-CoV-2 were found in dromedary camels that were MERS-CoV seropositive but MERS-CoV free, indicating that camels immunized with SARS-CoV-2 S protein could potentially be a prominent source of therapeutic agents for the prevention and treatment of COVID-19 [91]. A number of sdAbs have been identified from immunized camels, such as VHH-72 [39], C5, F2, H3, C1, [72] Nb15, Nb17, Nb19, Nb56, [15] Nb20, Nb21, [58] NIH-CoVnb-112 [59], aRBD-2, aRBD-5, aRBD-7, [68] S1-48, S1-RBD-6, [54] Nb15 [23], Fu2 [53], Ty1 [92], Nb11-59 [62], VHH E, VHH U, VHH V, and VHH W [44] (Table 2). All these sdAbs showed encouraging neutralizing activity against SARS-CoV-2 S pseudovirus or authentic virus and high binding affinity to RBD. Meanwhile, there are also several sdAbs with potent neutralizing activity and binding affinity that were isolated from non-immunized camels [41,63,70,74], fully human VHs library [25,45,71], or synthetic nanobody library [52,66,76] (Table 2). Several sdAbs were engineered to be multivalent or multispecific to improve binding affinity and breadth, neutralizing potency, therapeutic and prophylactic efficacy, and serum half-life [23,30,45,68,69,71,72,74,76,93]. For example, Nb15-Nb_H_-Nb15, consisting of one Nb (Nb_H_) specific for human serum albumin to prolong the half-life and two Nb15 specifics for RBD with (GGGGS)_3_ linker between each Nb, exhibited a more potent neutralization of the pseudotyped virus with an IC_50_ of 0.4 ng/mL (9.0 pM) and sustained in the lung for more than 168 h via intranasal administration (Table 2). Nb15-Nb_H_-Nb15 demonstrated both prophylactic and therapeutic efficacy against SARS-CoV-2 challenge in hACE2 transgenic mice at 10 mg/kg by intranasal administration [23]. In another study, Pittsburgh inhalable Nanobody 21 (PiN-21), a homotrimeric Nb21 fused with GS linker, could efficiently block SARS-CoV-2 infectivity at below 0.1 ng/mL in vitro and showed high therapeutic efficacy against SARS-CoV-2 infection in Syrian hamsters at ~0.2 mg/kg administrated by aerosolization, indicating that sdAbs administrated by inhalation could be an effective therapeutic strategy for treating COVID-19 infection [30,73] (Table 2). Furthermore, bn03, constructed by fusion of two sdAbs binding to two different highly conserved regions of RBD with four GGGGS linkers, exhibited therapeutic efficacy in hACE2 transgenic mice via inhalation administration and broadly neutralized all the SARS-CoV-2 variants [25] (Table 2). These findings demonstrated that sdAbs could be potential effective therapeutics for CoVs that are constantly mutating, such as SARS-CoV-2.

## 5. Influenza Viruses

Influenza viruses are negative-sense single-stranded segmented enveloped RNA viruses belonging to the *Orthomyxoviridae* family and become the most common causes of human respiratory infections with high morbidity and mortality [94]. There are three genera that could infect humans: influenza A virus (IAV), influenza B virus (IBV), and influenza C virus (ICV) (Table 1). The majority of seasonal influenza viruses are IAVs and IBVs [95], which contain eight RNA segments that encode eight proteins, including two major targets of neutralizing antibodies, HA (a homotrimer that mediates the viral entry process) and NA (critical for viral release) [96]. IAVs are classified into subtypes based on the combination of HA and NA. Furthermore, these subtypes are classified into group 1 (H1, H2, H5, H6, H8, H9, H11, H12, H13, H16, H17) and group 2 (H3, H4, H7, H10, H14, H15), according to the phylogenetics of HA [97]. Due to the high genome mutation rate and the existence of antigenic drift and shift, new IAV types constantly appear and escape from the currently available neutralizing antibodies of influenza viruses [98]. Therefore, it is necessary to develop universal antibodies recognizing the conserved sites of HA or NA.

The monomer HA contains two subunits, HA1 and HA2. The head domain of the HA1 subunit is antigenic and variable, resulting in specific neutralizing of one influenza virus type for several sdAbs recognizing this region [40,77,78]. However, the receptor-binding site (RBS) is relatively conserved and has been recognized as the main target of broadly neutralizing antibodies. In contrast, the HA2 mediated membrane fusion shows more conserved [99], and, and the majority of broadly neutralizing sdAbs recognize the stem region [24,80]. Laursen et al. reported four sdAbs isolated from the immunized llamas, among which SD36, SD38, and SD83 bound to the conserved stem region of HA, while SD84 bound to the relatively conserved region of the HA head domain. SD36 and SD38 exhibited exquisite neutralizing breadth of IAVs, while SD83 and SD84 potently neutralized IBVs (Table 2). Moreover, the multidomain antibody was engineered by genetically fusing individual sdAbs with peptide linkers and then linking to human IgG1-Fc, named MD3606, exhibited enhanced broadly neutralizing activity against IAVs and IBVs and superior protective activity in mice from influenza B infection [24]. Further, humanized MD3606 expressed using an adeno-associated virus vector showed potent prophylactic efficacy in mice challenged with H1N1, H3N2, or IBV at a dose of 4 × 10^7^ to 5 × 10^9^ genome copies per mouse by intranasal administration [24]. In another study, R1a-B6 binding to the stem region of HA with high affinity was developed from an immunized alpaca by phage display and demonstrated neutralizing breadth against H1N1, H5N1, and H9N2 viruses. The bivalent R1a-B6 showed more neutralizing strength, including the H2N2 virus [80,82] (Table 2). Moreover, R1a-B6-Fc fusion protein delivered by AAV could be sustained in sera for more than 6 months and protected mice challenged with lethal H1N1 or H5N1 viruses [81]. In contrast, some sdAbs have been reported that bound to the HA stem region; however, they were only effective for a specific strain of influenza virus [78,79].

NA is a tetramer with each monomer consisting of four distinct domains: the catalytic head, the stalk region, the transmembrane region, and the cytoplasmic tail [100]. Due to the low mutation rate of HA, some universal mAbs targeting NA have been isolated from infected or vaccinated people, demonstrating NA as a promising target for broad protection against various influenza viruses [101]. N1-3-VHHb, a bivalent nanobody isolated from immunized alpaca and binding to H5N1 NA, showed potent NA-inhibiting activity and antiviral potential in vitro, as well as prophylactic efficacy in mice, challenged with lethal H5N1 virus administered intranasally at 60 μg [46].

M2 is an ion channel protein activated by the low pH of the endosome and plays essential roles in releasing the viral genome from the endosome to the cytoplasm [43]. The 1–9, His37, and Trp41 residues of M2 in all IAVs are extremely conserved. Therefore, sdAbs targeting these residues of M2 showed universal protection against IAVs [102]. Wei et al. reported a sdAb M2-7A that bound to the recombinant M2 protein and the native M2 on the virion, as well as the M2, expressed on the cell surface, but not to the synthetic ectodomain of M2 (M2e, residues 1–24) (Table 2). M2-7A broadly inhibited the replication of H3N2 and H1N1 in vitro and protected mice challenged with lethal H1N1 virus in vivo [42]. In addition, some VHHs targeting M2e showed broad binding breadth to most H2N2, H1N1 and H3N2 avian and swine influenza viruses. One of these VHHs, named M2e-VHH-23m, was fused with FcγRIV by a (GGGGS)_3_ linker and exhibited potently prophylactic efficacy in mice challenged with lethal H3N2 virus at 50 μg [83]. Furthermore, an mRNA treatment that encoded this bispecific sdAb also showed protection against the same H3N2 virus strain [84] (Table 2).

Because of the relatively high mutation rate of HA, NA, and M2, the development of truly universal antibodies remains challenging. To overcome this challenge, researchers focus on the less variable proteins, such as the nucleoprotein coating the viral RNA that is critical to transporting viral ribonucleoproteins (vRNPs) into the nucleus. More than 20 sdAbs targeting the nucleoprotein have been identified [47,103,104]. One of them, named αNP-VHH1, exhibited potent antiviral activity by blocking vRNP nuclear import and subsequent viral transcription and replication [47,105] (Table 2). In addition, Schmidt et al. reported a lentiviral screening strategy that developed several nucleoprotein-specific VHHs showing IAV-inhibiting activity by blocking the nuclear import of vRNPs and viral mRNA transcription [104].

## 6. Respiratory Syncytial Virus

Respiratory syncytial virus (RSV) is the main cause of respiratory virus infections in young children and elderly people. Apart from a humanized mAb that is licensed for high-risk infants as passive immunoprophylaxis, there are no effective RSV vaccines [38]. RSV is a non-segmented, negative-sense, single-stranded RNA virus belonging to the *Pneumoviridae* family (Table 1). Based on the antigenic subgroups, RSV is classified into two types: RSV-A and RSV-B. RSV is a filamentous enveloped virus and its envelope contains three proteins on the surface: the glycoprotein that plays an important role in host cell attachment and is the most variable structural protein, the fusion protein (F) that mediates the process of fusion and cell entry, and the small hydrophobic protein that is not required for the entry process [38,106]. The sequences of the F ectodomains share a high similarity (~90%) between RSV-A and RSV-B. Therefore, most sdAbs were developed for RSV F protein. ALX-0171, the most intensively studied nanobody targeting RSV F, is a trimeric nanobody developed by Ablynx with two GS linkers among each subunit (Table 2). ALX-0171 demonstrated high binding affinity with the antigenic site II epitope of F and broadly neutralizing activity against different clinical RSV isolates in vitro, as well as therapeutic efficacy in cotton rats challenged with RSV Tracy by nebulization administration [14]. In the preclinical study, ALX-0171 also exhibited robust antiviral effects and safety in newborn lambs infected with hRSV-M37 [29]. The phase I/IIa clinical trial showed that ALX-0171 administrated by inhalation at 1.2 mg/kg in infants infected with RSV could reduce the nasal viral titters with no safety concerns (https://www.clinicaltrialsregister.eu/ctr-search/trial/2014-002841-23/results, accessed on 18 August 2016). However, ALX-0171 showed no significant effect in children who were hospitalized with lower respiratory RSV infection in the phase IIb clinical trial [107]. With extensive research on the structure and function of RSV surface F glycoproteins, a number of sdAbs specifically targeting the prefusion conformation of F protein (PreF) exhibited higher binding affinity and inhibitory activity against RSV in vitro [85,86,87]. For example, F-VHH-4 and F-VHH-L66 could bind to PreF with <18 pM and 154 pM, respectively, and broadly neutralized RSV in vitro (Table 2). The crystal structures of F-VHH-4 and -L66 showed that they were both bound to a cavity formed by the boundary of two F protomers. In addition, F-VHH-4 demonstrated better prophylactic efficacy than palivizumab in mice challenged with RSV A2 at 10 μg [86]. These PreF-binding sdAbs with picomolar affinity are potential therapeutics for RSV infections.

## 7. Other Respiratory RNA Viruses

In addition to CoVs, influenza viruses, and RSV, there are other respiratory RNA viruses, such as HRVs, human metapneumoviruses (hMPVs), and parainfluenza viruses (PIVs). HRVs, usually associated with upper respiratory tract infection, are with positive-sense, single-stranded RNA genome and belong to the *Picornaviridae* family (Table 1). They are genetically classified into three types (RV-A, RV-B, and RV-C) [33]. The capsid of HRVs consists of four proteins, VP1, VP2, VP3, and VP4. Except for VP4, VP1, VP2, and VP3 are exposed at the viral surface and therefore are the main targets for neutralizing antibodies. The canyon in VP1 is the attachment site for host cell receptors. For most known HRV serotypes (>90%), the intercellular adhesion molecule 1 (ICAM-1) acts as their receptor for viral entry, whereas a small portion of HRVs enter the host cell via the low-density lipoprotein receptor (LDLR) [33,108]. A number of mAbs and Fabs targeting the VP1 [109], VP3 [110], or ICAM-1 [111] have been identified with neutralizing activity, while there are no available sdAbs targeting HRVs. PIVs are the second leading cause of acute respiratory tract infections in children. They are negative-sense single-stranded enveloped RNA viruses belonging to the *Paramyxoviridae* family [35,112] (Table 1). Based on the genetic diversity, PIVs can be divided into four serotypes, PIV-1, PIV-2, PIV-3, and PIV-4. There are two glycoproteins studded in the viral envelope. One of them is the HA-NA (HN) protein, which serves as the attachment protein by binding to the sialic acid-containing molecules of the host cells. The other one is the F protein responsible for the fusion. These two proteins are exposed on the viral surface and are the main targets for neutralizing antibodies and prophylactic vaccines [112]. There are no effective vaccines or therapeutics for PIV infections currently, and only several antibodies targeting the apex of F protein were developed and exhibited protective activity in cotton rats [113]. hMPVs, a common cause of viral pneumonia among infants and children, are enveloped viruses with negative-sense single-stranded RNA genomes and belong to the *Pneumoviridae* family [34] (Table 1). Similar to RSVs, there are three proteins exposed on the hMPV surface, the glycoprotein, the F protein, and the small hydrophobic protein. Among them, the F protein could induce neutralizing antibodies and hence is the target for developing neutralizing antibodies or vaccines [114].

## 8. Conclusions and Perspectives

In the past twenty years, respiratory RNA virus infections caused several outbreaks, with high morbidity and mortality [115,116,117]. The main characteristics of respiratory RNA viruses are strong transmission capacity and a high mutation rate. For instance, the current pandemic of SARS-CoV-2 has lasted for more than two years and five variants of concern (VOCs), including Alpha, Beta, Gamma, Delta, and Omicron, have been identified. The latest mutant strain Omicron has been reported to be resistant to most neutralizing antibodies and currently available vaccines [118,119]. Because of these characteristics, the developments of universal vaccines and effective therapeutics specific for respiratory RNA viruses remain a great challenge.

mAb-based therapeutics for respiratory RNA virus infections have been demonstrated to be effective [120]. However, the development of mAbs against viral infections has been hampered due to expensive production costs and limited commercial markets. Since the end of the last century, sdAbs with smaller size, higher stability and solubility, and lower immunogenicity and production costs have attracted more and more attention [18]. Compared to mAbs, sdAbs have advantages that have been confirmed by many studies. Firstly, sdAb can penetrate deep inside the “sterically hidden” interface of the virus and neutralize the virus. As the enveloped virus with flexible viral states, the large size of mAbs might bind the cryptic epitope in one state; however, it exhibited neglectable neutralization. Importantly, these cryptic domains are usually highly conserved and induce a low immune response in humans. Secondly, sdAb can be easily engineered to be multivalent to improve neutralizing potency and breadth [22]. Notably, by inhaled delivery, sdAbs can be directly delivered to the lung as the main infectious tissue in respiratory diseases [25]. Therefore, these sdAbs are promising to develop universal antiviral therapeutics for respiratory RNA virus infections and provide insights into the rational design of effective vaccines.

Despite the superior properties of sdAbs, mAbs are still dominant in the treatment of viral infections. For instance, 6 mAbs have been FDA approved for COVID-19 therapy and more than 50 ongoing clinical trials. The short half-life in vivo may be the limitation of sdAb developments. Currently, many strategies have been used to extend the half-life of sdAbs, such as fusing antiviral sdAbs with anti-human serum albumin nanobodies or IgG1 Fc fragments [121,122,123]. So far, only one sdAb has been approved for treating aTTP, and more than 30 sdAb-based drug candidates are in clinical trials. We expect recent technological advances in the fields of half-life extension, nebulized delivery, and production processes to advance more sdAbs into the clinic for the treatment of respiratory diseases.

## Figures and Tables

**Figure 1 viruses-14-01162-f001:**
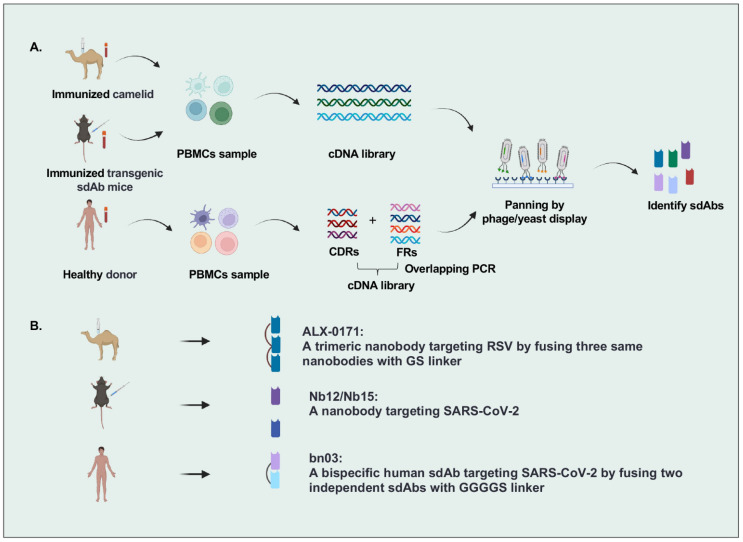
Overview of single-domain antibodies screening strategies. (**A**). Strategies to identify single-domain antibodies. In brief, VHH or VH genes are cloned from peripheral blood lymphocytes of specific-antigen immunized camelids, transgenic sdAb mice, or healthy donors, and then antigen-specific sdAbs are identified by phage/yeast display panning. (**B**). Representative single-domain antibodies against respiratory RNA viruses are identified by different strategies. Figure generated with BioRender.

**Figure 2 viruses-14-01162-f002:**
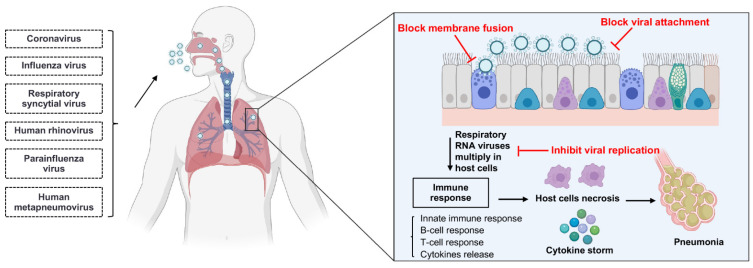
The mechanisms of single-domain antibodies inhibiting respiratory RNA virus infections. Single-domain antibodies inhibit respiratory RNA virus infections by several different mechanisms in parallel with the virus life cycle. Respiratory RNA viruses primarily inoculate through the nose and then enter into host cells. After entry, respiratory RNA viruses multiply in the host cells, such as epithelium cells of the large and small airways, vascular endothelial cells, and alveolar macrophages. The infection of the respiratory tract can induce an immune response, and the immune response may lead to the cytokine storm; finally, the inflammation and necrosis of the epithelium cells may lead to pneumonia. The majority of neutralizing sdAbs inhibit the entry process by blocking the attachment of membrane fusion between virus and host cell. Moreover, some sdAbs neutralize the virus by inhibiting the viral genomic replication in host cells and the release of virion progeny from the host cells. Figure generated with BioRender.

**Table 1 viruses-14-01162-t001:** The characteristics of respiratory RNA virus.

Respiratory RNA Virus	Genome	Family	Envelope	Potential Viral Targets for Developing sdAbs
Coronavirus	Positive-sense, single-stranded RNA genome	* Coronaviridae *	Enveloped	S protein (RBD)
Influenza virus	Negative-sense, single-stranded, segmented RNA genome	* Orthomyxoviridae *	Enveloped	HA, NA, M2, NP
Respiratory syncytial virus	Negative-sense, single-stranded, non-segmented RNA genome	* Pneumoviridae *	Enveloped	F protein
Human rhinovirus	Positive-sense, single-stranded RNA genome	* Picornaviridae *	Non-enveloped	VP1, VP2, VP3
Parainfluenza virus	Negative-sense, single-stranded RNA genome	*Paramyxoviridae*	Enveloped	HN, F protein
Human metapneumovirus	Negative-sense, single-stranded, non-segmented RNA genome	*Pneumoviridae*	Enveloped	F protein

**Table 2 viruses-14-01162-t002:** Single-domain antibodies for respiratory RNA virus.

Respiratory RNA Virus	sdAbs	Form	Origin	Binding Region	Affinity (K_D_)	Neutralizing Activity (IC_50_)	Antiviral Activity in Animal Model	Reference
SARS-CoV-1	VHH-72	Monomer	Immunized llama	RBD	1.15 nM	N/A	N/A	[39]
	S1-1, S1-RBD-6, S1-39	Monomer	Immunized llama	RBD	N/A	Pseudoviruses: 8.6 nM, 89.7 nM, 22.1 nM	N/A	[54]
	S14	Monomer	Immunized alpaca	RBD	0.143 nM	Pseudoviruses: 4.93 ng/mL	N/A	[55]
	5F8	Monomer	Synthetic humanized nanobody library	RBD	239.2 nM	N/A	N/A	[52]
MERS-CoV	VHH-83	Monomer	Immunized dromedary camel	RBD	0.103 ± 0.141 nM	Authentic viruses: 30 pM (PRNT50)	Prophylactic in K18 transgenic mouse at 200 μg pre mouse	[56]
	NbMS10	Monomer	Immunized llama	RBD	0.87 nM	Pseudoviruses: 3.52 μg/mL	Prophylactic and therapeutic in hDPP4-Tg mice at 10 mg/kg	[57]
	VHH-55	Monomer	Immunized llama	RBD	0.079 nM	Pseudoviruses: 0.014 µg/mL	N/A	[39]
SARS-CoV-2	VHH-72	Monomer	Immunized llama	RBD-SD1	38.6 nM	Pseudoviruses: 0.2 µg/mL	N/A	[39]
	Nb21	Monomer	Immunized llama	RBD	<1 pM	Pseudoviruses: 0.0495 nM Authentic viruses: 0.022 n	N/A	[30,58]
	Nb12, Nb30	Monomer	Immunized nanomouse	RBD	30 nM, 6.55 nM	Pseudoviruses: 11.7 nM, 6.9 nM	N/A	[15]
	Nb15, Nb17, Nb19, Nb56	Monomer	Immunized llama	RBD	8.15 nM, 5.59 nM, 4.72 nM, 3.26 nM	Pseudoviruses: 0.4 nM, 0.6 nM, 0.3 nM, 0.9 nM	N/A	[15]
	NIH-CoVnb-112	Monomer	Immunized llama	RBD	4.94 nM	Pseudoviruses: 0.323 µg/mL (23.1 nM)	Prophylactic and therapeutic in a hamster model administrated via nebulization	[59,60]
	Re9F06	Monomer	Immunized alpaca	RBD	4 nM	Authentic viruses with D614G mutation: 17 nM	N/A	[61]
	Nb11-59	Monomer	Immunized camel	RBD	21.6 nM	Authentic viruses: 0.55 μg/mL	N/A	[62]
	VHH E, VHH U, VHH V, VHH W	Monomer	Immunized camel	RBD	1.86 nM, 21.4 nM, 8.92 nM, 22.2 nM	Pseudoviruses: 60 nM, 286 nM, 198 nM, 257 nM Authentic viruses: 48 nM, 185 nM, 142 nM, 81 nM	N/A	[44]
	H11-H4, H11-D4	Monomer	Non-immunized llama	RBD	12 ± 1.5 nM, 39 ± 2 nM	(Fused with Fc of human IgG1) Authentic viruses: 6 nM, 18 nM	N/A	[41]
	MR3	Monomer	Non-immunized camels	RBM	1.0 nM	Pseudoviruses: 0.42 μg/mL	Prophylactic in hamster model at 2.5 mg of divalent MR3	[63]
	7A3	Monomer	Non-immunized camels	RBD	0.2 nM	N/A	Prophylactic in K18-hACE2 mice at 10 mg/kg	[64]
	N3113	Monomer	Fully human VHs library	RBD	57.01 ± 1.52 nM	Pseudoviruses:18.9 μg/mL	N/A	[16]
	n3113.1	Monomer	Fully human VHs library	RBD	63.8 nM	Pseudoviruses:6 μg/mL	Prophylactic and therapeutic in a hACE2-transgenic mouse model at 40 mg/kg	[45]
	K-874A	Monomer	Synthetic VHH-cDNA display	S1	1.4 nM	Authentic viruses: 5.74 ± 2.6 μg/mL	Prophylactic and therapeutic in the Syrian hamster model at 30 mg/kg	[65]
	Sb23	Monomer	Synthetic nanobody library	RBD	10.6 ± 2.0 nM	Pseudoviruses: 0.6 μg/mL	N/A	[66]
	1E2, 2F2, 3F11, 4D8, 5F8	Monomer	Synthetic humanized nanobody library	RBD	35.52 nM, 5.175 nM, 3.349 nM, 6.028 nM, 0.996 nM	Pseudoviruses: 5.324 μg/mL, 0.742 μg/mL, 0.066 μg/mL, 0.781 μg/mL, 0.072 μg/mL Authentic viruses: 18.47 μg/mL, 22.62 μg/mL, 28.64 μg/mL, 9.628 μg/mL, 39.28 μg/mL	N/A	[52]
	SR6v15	Monomer	Cell-free Library (nanobody, non-immune)	RBD	2.18 nM	Pseudoviruses: 3.591 ± 0.043 nM	N/A	[67]
	aRBD-2-5, aRBD-2-7	Dimer	Immunized alpaca	RBD	0.0592 nM, 0.252 nM	Authentic viruses: 0.00122 µg/mL (0.043 nM), 0.00318 µg/mL (0.111 nM)	N/A	[68]
	Nb91-Nb3-hFc	Dimer	non-immunized Bactrian camel	RBD	N/A	Pseudoviruses: 1.54 nM	N/A	[69]
	Nanosota-AC-Fc	Dimer	non-immunized llamas and alpacas	RBD	15.7 pM	Pseudoviruses: 0.27 μg/mL	N/A	[70]
	bn03	Dimer	Fully human VHs library	RBD	<1 nM	Pseudoviruses: 0.11–0.76 μg/mL	Therapeutic in hACE2-transgenic mice model administrated by inhalation	[25]
	KC3.ep3 (Fc-fusion proteins)	Dimer	Synthetic nanobody library	RBD	34 ± 1 pM	Pseudoviruses: 1.82 ± 1.09 ng/mL, Authentic Viruses: 38.53 ± 3.98 ng/mL	N/A	[71]
	Nb15-NbH-Nb15	Trimer	Immunized Alpaca	RBD	0.541 nM	Pseudoviruses: 0.4 ng/mL	Prophylactic and therapeutic in hACE2 transgenic mice at 10 mg/kg via intranasal administration	[23]
	C5-trimer	Trimer	Immunized Llama	RBD	18 pM	Authentic Viruses: 18 pM (Victoria-B), 25 pM (Alpha- B1.1.7)	Prophylactic and therapeutic in the Syrian hamster model of COVID-19 administrated via the intranasal route	[72]
	PiN-21	Trimer	Immunized llama	RBD	N/A	Pseudoviruses: 1.321 pM Authentic viruses: 6.039 pM	Prophylactic and therapeutic in Syrian hamsters administrated via nebulization	[30,73]
	3F-1B-2A-Fc	Trimer	Non-immunized llama	RBD	0.0468 nM	Pseudoviruses: 6.44 nM	N/A	[74,75]
	Nb6-tri	Trimer	Synthetic nanobody library	RBD	41 nM	Pseudoviruses: 1.2 nM Authentic viruses: 140 nM	N/A	[76]
Influenza virus	NB7-14	Monomer	Immunized alpaca	H7 head region of HA	2.63 nM	Pseudoviruses: 17 nM	N/A	[40]
	aHA-7	Trimer	Immunized camel	H5 head region of HA	N/A	Authentic viruses: 4.2 nM	Prophylactic in mice infected with lethal H5N2 virus administered intranasally at 50 μg pre mouse	[77]
	Vic 2a-6	Monomer	Immunized alpaca	IBV head region of HA	0.08 nM	Pseudoviruses: 10 nM	N/A	[78]
	Vic 1b-10	Monomer	Immunized alpaca	IBV stem region of HA	0.28 nM	Pseudoviruses: 0.2 nM	N/A	[78]
	2F2, H1.2, G2.3	Monomer	Immunized alpaca	H1 stem domain of HA	15.7 nM, 3.65 nM, 5.54 nM	N/A	N/A	[79]
	R1a-B6	Monomer	Immunized alpaca	H1, H5, H9 stem regions of HA	0.86 nM (H1), 0.52 nM (H5), 6.89 nM (H9)	Authentic viruses: 3.2 ± 0.5 nM (H1), 5.5 ± 0.9 nM (H5), 182.2 ± 25.2 nM (H9)	Prophylactic in mice infected with lethal influenza virus by AAV-Mediated delivery	[80,81,82]
	SD36	Monomer	Immunized llama	Influenza A group 2 virus stem region of HA	N/A	Pseudoviruses: 10–100 nM	N/A	[24]
	SD38	Monomer	Immunized llama	Influenza A group 1 virus stem region of HA	N/A	Pseudovirus: 10–100 nM	N/A	[24]
	SD38-SD36	Dimer	Immunized llama	IAV stem region of HA	N/A	Pseudovirus: 1–100 nM	N/A	[24]
	SD83-SD84	Dimer	Immunized llama	IBV stem region of HA	N/A	Pseudovirus: 1–100 nM	N/A	[24]
	N1-3-VHHb	Dimer	Immunized alpaca	H5 NA	0.37 nM	Authentic viruses: 7.6 ± 3.0 nM	Prophylactic in mice infected with lethal H5N1 virus administered intranasally at 60 μg pre mouse	[46]
	M2-7A	Monomer	Synthetic camel VHH libraries	M2	39.5 nM	N/A	Therapeutic in mice infected with lethal H1N1 virus at 200 μg pre mouse	[42]
	FcgRIV VHH-M2e VHH	Dimer	Immunized llama	M2e	729.9 nM	N/A	Prophylactic and therapeutic in mice infected with lethal H3N2 virus at 50 μg pre mouse	[83,84]
	αNP-VHH1	Monomer	Immunized alpaca	NP	N/A	N/A	N/A	[47]
Respiratory syncytial virus	m17, m35	Monomer	Human antibody VH library	Prefusion F	0.544 nM, 0.386 nM	Authentic viruses: 0.67 μg/mL, 0.70 μg/mL	N/A	[85]
	F-VHH-4, F-VHH-L66	Monomer	Immunized llama	Prefusion F	<18 pM, 154 pM	Authentic viruses: <0.1 nM	Prophylactic in mice challenged with RSV at 30 μg pre mouse	[86]
	F-VHH-Cl184	Monomer	Immunized llama	Prefusion F	84 pM	Authentic viruses: 0.4 nM (RSV-A), 21.6 nM (RSV-B)	N/A	[87]
	F-VHHb	Dimer	Immunized llama	F	1.78 nM	Authentic virus: 0.14 nM (RSV-A), 103 nM (RSV-B)	Prophylactic and therapeutic in mice infected with RSV at 60 μg pre mouse	[88,89]
	ALX-0171	Trimer	Immunized llama	F	0.113 nM	Authentic virus: 0.1 ± 0.07 nM (RSV-A), 0.4 ± 0.2 nM (RSV-B)	Therapeutic in cotton rats challenged with RSV Tracy by inhalation	[14]

N/A: not available.

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
