# Peer review of "Single-Domain Antibodies as Therapeutics for Respiratory RNA Virus Infections"

_viruses, 2022, doi:10.3390/v14061162_

Round 1

Reviewer 1 Report

In my opinion this is a very interesting and well-written review that can be published in its current form. It contains a very deep description of the availbale antiviral sdAb and discusses their potential use as therapeutics for respiratory RNA viral infections. It alo describes the briefly how these antibodies can be generated and what are mechanisms of action. All in all very good written and easy to follow.I highly recommend it for publication.

Minor comment:

Title in Table: please delete the dot;

Author Response

In my opinion this is a very interesting and well-written review that can be published in its current form. It contains a very deep description of the available antiviral sdAb and discusses their potential use as therapeutics for respiratory RNA viral infections. It also describes the briefly how these antibodies can be generated and what are mechanisms of action. All in all very good written and easy to follow. I highly recommend it for publication.

Minor comment:

Title in Table: please delete the dot;

Response: We thank Reviewer 1 for the recognition of our review. As for the comment, we have deleted the dot in the title of Table 1 and Table 2.

Reviewer 2 Report

Manuscript ID: viruses-1736848

This manuscript describes an extended review on Single-domain antibodies as therapeutics for respiratory RNA virus infections. Different studies were categorized about a specific class of antibodies, named nanobodies or single-domain antibodies (sdAb) derived from camelids or synthetically engineered, which were used as therapeutics vaccination strategies against Coronavirus, Influenza and Respiratory Syncytial virus infections. Especially the current COVID-19 pandemic and the need for new vaccination agents/antivirals against SARS-CoV-2 has clearly set the spotlight on these therapeutics. The review is acceptable for publication.

Some minor points:

  • In ln 117-118 it is mentioned that “…some studies have reported that directly delivering nanobodies into the lungs can block the viral invasion of airway epithelial cells in situ by inhalation” without referring those studies. Please add the references.
  • In ln 148 the abbreviation ACE2 is mentioned for the first time without the full name description. This was done later in the text in ln 176. Please switch.
  • In ln 203 the authors mentioned “ND50 of NbMS10 was 3.52 µg/ml.” This must be changed into IC50 as this is also mentioned in the Neutralizing activity (IC50) column in Table 2.
  • In Table 2 under Influenza virus, the aNP-VHH1 sdAb was referred as reference [45]. In ln 311 reference [93], is mentioned. Is this correct?

Author Response

This manuscript describes an extended review on Single-domain antibodies as therapeutics for respiratory RNA virus infections. Different studies were categorized about a specific class of antibodies, named nanobodies or single-domain antibodies (sdAb) derived from camelids or synthetically engineered, which were used as therapeutics vaccination strategies against Coronavirus, Influenza and Respiratory Syncytial virus infections. Especially the current COVID-19 pandemic and the need for new vaccination agents/antivirals against SARS-CoV-2 has clearly set the spotlight on these therapeutics. The review is acceptable for publication.

Response: We thank the reviewer 2 for recognition of our review and greatly appreciate the suggestions given by the reviewer. We revised our review based on these suggestions.

ln 117-118 it is mentioned that “…some studies have reported that directly delivering nanobodies into the lungs can block the viral invasion of airway epithelial cells in situ by inhalation” without referring those studies. Please add the references.

Response: We thank the reviewer for this suggestion. We have added the references for this sentence in revised manuscript and marked by yellow.

In 148 the abbreviation ACE2 is mentioned for the first time without the full name description. This was done later in the text in ln 176. Please switch.

Response: We thank the reviewer for the detailed suggestion. We have spelled out the full name description of ACE2 at first mention, and replaced with abbreviation in the later text.

In ln 203 the authors mentioned “ND50 of NbMS10 was 3.52 µg/ml.” This must be changed into IC50 as this is also mentioned in the Neutralizing activity (IC50) column in Table 2.

Response: We thank the reviewer for the detailed suggestion. We have replaced the “ND50” with “IC50”.

In Table 2 under Influenza virus, the aNP-VHH1 sdAb was referred as reference [45]. In ln 311 reference [93], is mentioned. Is this correct?

Response: We thank the reviewer for this suggestion. The reference cited in Table 2 is the original paper that identified the sdAb (αNP-VHH1) against influenza virus. In contrast, the reference [93] indicated the antiviral mechanism of αNP-VHH1 by determining the crystal structure of NP/ αNP-VHH1. Now, the two references have been cited in the revised text.

Reviewer 3 Report

In this manuscript, Huang and colleagues summarize the latest developments in single domain antibody research with respect to their potential application as therapeutics for respiratory RNA viral diseases

Overall, this is a great summary of the recent areas of research as well as advances in the field. I found the review informative and the sections broken down and discussed in a logical manner. I have no major issues with the manuscript in general. However, there are numerous minor spelling/ grammar and formatting inconsistencies need to be addressed.

For instance:

Line 125: The phrase “RNA viruses can be classified into enveloped and non-enveloped viruses based on key structural distinction in lipid membrane.” Is ambiguous. Perhaps saying “ enveloped or non-envelopeloped viruses based on the presence or absence of lipid membrane” would be more clear.

Table 1 lists “The characteristics of respiratory RNA virus.” It is unclear whether the targets listed are previously shown or just potential protein targets for each virus? A more specific caption would be helpful. Further, it is unclear why the font size is inconsistent in this section.

 Line 149: Inconsistent tense. Please correct.

Line 163: It is unclear what the authors wish to convey. If the viruses cause mild to moderate disease, how can they cause outbreaks of deadly respiratory disease? Can the authors rephrase to clarify this sentence?

Table 2 has been inserted in the document twice. Please correct the duplication. Further, can these tables be presented in landscape layout, rather than portrait layout, this will improve usability of the tables significantly.

 Line 246: Saying “There are three genera” would be more appropriate here according to ICTV rather that “There are three types”

Line 307: It should read “as the nucleoprotein coating the viral RNA ” instead of “as the nucleoprotein coating the viral ribonucleoprotein (vRNP)" since vRNP is vRNA+NP.

Line 338: Needs to be corrected: “With extensive researches for structure and function of RSV surface F glycoproteins of RSV”

Overall, a thorough round of review by the authors for language and formatting errors, will improve the clarity and utility for readers.

Author Response

In this manuscript, Huang and colleagues summarize the latest developments in single domain antibody research with respect to their potential application as therapeutics for respiratory RNA viral diseases

Overall, this is a great summary of the recent areas of research as well as advances in the field. I found the review informative and the sections broken down and discussed in a logical manner. I have no major issues with the manuscript in general. However, there are numerous minor spelling/ grammar and formatting inconsistencies need to be addressed.

Response: We thank Reviewer for the insightful and helpful suggestion herein to improve our manuscript. We have addressed these issues in revised text. Additionally, we consulted a professional editing service and asked a colleague who is native speaker to check the English. Thanks for this comment.

Line 125: The phrase “RNA viruses can be classified into enveloped and non-enveloped viruses based on key structural distinction in lipid membrane.” Is ambiguous. Perhaps saying “enveloped or non-enveloped viruses based on the presence or absence of lipid membrane” would be more clear.

Response: We thank for this detailed suggestion. We have revised this sentence into “RNA viruses can be classified into enveloped and non-enveloped viruses based on the presence or absence of lipid membrane”.

Table 1 lists “The characteristics of respiratory RNA virus.” It is unclear whether the targets listed are previously shown or just potential protein targets for each virus? A more specific caption would be helpful. Further, it is unclear why the font size is inconsistent in this section.

Response: We thank for the detailed review and helpful, constructive comments on our manuscript. In Table 1, the column “Targets for sdAbs” summarized the viral proteins as known and potential targets for developing neutralizing antibodies. To clarify more clearly and specific, we have modified this caption to “Potential viral targets for developing sdAbs”.

Line 149: Inconsistent tense. Please correct.

Response: We thank for this detailed suggestion. We have corrected it.

Line 163: It is unclear what the authors wish to convey. If the viruses cause mild to moderate disease, how can they cause outbreaks of deadly respiratory disease? Can the authors rephrase to clarify this sentence?

Response: We thank for this comment. We have modified this sentence in the revised text, as shown in the quoted text.

“Coronaviruses (CoVs) are enveloped viruses with a positive-sense RNA genome that usually cause mild to moderate respiratory diseases in humans, which lead to several outbreaks of respiratory diseases.”

Table 2 has been inserted in the document twice. Please correct the duplication. Further, can these tables be presented in landscape layout, rather than portrait layout, this will improve usability of the tables significantly.

Response: We thank for this suggestion. We have deleted one “Table 2”. Moreover, we have modified the page layout of Table 2, and now the Table 2 is presented in landscape layout.

Line 246: Saying “There are three genera” would be more appropriate here according to ICTV rather that “There are three types”

Response: We greatly appreciate this comment. We have changed “types” into “genera”.

Line 307: It should read “as the nucleoprotein coating the viral RNA ” instead of “as the nucleoprotein coating the viral ribonucleoprotein (vRNP)" since vRNP is vRNA+NP.

Response: We really appreciate the comment for improving our review. We have revised this sentence (as in the text below).

“In order to overcome this challenge, researchers focus on the less variable proteins, such as the nucleoprotein coating the viral RNA that is critical to transport viral ribonucleo-proteins (vRNPs) into the nucleus.”

Line 338: Needs to be corrected: “With extensive researches for structure and function of RSV surface F glycoproteins of RSV”

Response: We thank for this detailed suggestion. We have revised this sentence and deleted the final two words “of RSV”.

Round 2

Reviewer 3 Report

All my comments have been addressed adequately.